# Model Simplification for Asymmetric Marine Vehicles in Horizontal Motion—Verification of Selected Tracking Control Algorithms

Przemyslaw Herman 

Institute of Automatic Control and Robotics, Poznan University of Technology, ul. Piotrowo 3a,
60-965 Poznan, Poland; przemyslaw.herman@put.poznan.pl; Tel.: +48-61-224-4500

**Abstract:** This paper addresses a trajectory tracking control algorithm for underactuated marine vehicles moving horizontally in which the current in the North–East–Down frame is constant. This algorithm is a modification of a control scheme based on the input-output feedback linearization method, for which the application condition was that the vehicle was symmetric with respect to the left and right sides. The proposed control scheme can be applied to a fully asymmetric model, and, therefore, the geometric center can be different from the center of mass in both the longitudinal and lateral directions. A velocity transformation to generalized vehicle equations of motion was used to develop a suitable controller. Theoretical considerations were supported by simulation tests performed for a model with 3 degrees of freedom, in which the performance of the proposed algorithm was compared with that of the original algorithm and the selected control scheme based on a combination of backstepping and integral sliding mode control approaches.

**Keywords:** underactuated underwater vehicle; nonlinear control; horizontal motion; trajectory tracking; simulation

## 1. Introduction

The issue of the control of marine vehicles, especially underwater or surface vehicles, has been of interest to researchers for many years. When designing such vehicles, various problems must be taken into account, such as propulsion limitations, achievable speeds, parameter uncertainty, external disturbances or the absence of at least one control signal. Among the implemented motion control tasks of various marine vehicles, tracking control is often encountered, which allows the system to achieve the desired trajectory with admissible performance.

From the viewpoint of underactuated marine surface vessel control (as well as for other vehicles moving horizontally), there are various important challenges. Some of them, for example, were mentioned in [1], including the following: (1) the vehicles belong to the nonholonomic class which cannot be directly transformed into a standard chain system because there is no relative degree; (2) when navigating, these types of vehicles inevitably experience internal and external uncertainties, including parameter uncertainties, nonparametric uncertainties and external disturbances, that result from the modeling techniques used and the unpredictable marine environment, making accurate model knowledge unavailable for control design. Despite these evident difficulties, it is common practice to use a dynamic model with a diagonal inertia matrix, which is equivalent to assuming that the center of mass lies at the same point as the geometric center.

Given the above, the motivation of this article is to try to analyze whether such a simplified vehicle model can be used when there are offsets of the center of mass in the longitudinal and lateral directions. Even more important is the search for such a scheme that guarantees acceptable control performance in the above case.

Tracking control based on a vehicle model with a diagonal inertia matrix has been discussed in the marine vehicle literature for many years. Only some solutions to this problem will be recalled here. A tracking controller based on the concept of a virtual ship in a ship body frame was presented in [2]. In order to provide robustness against inaccurately known parameters and external disturbances, sliding mode control (SMC) is very often used, as for example in [3,4]. Backstepping techniques and Lyapunov's direct methods are also applied to build control schemes [5–7]. However, in many cases, the combination of SMC with another method has also proven effective. In [8,9], a backstepping approach together with SMC, while in the integral SMC [10,11], were applied. A very common practice among control strategies is the introduction of neural networks (NN) along with other methods. For example, in [12], NN were used with Lyapunov-based theory, a high-gain observer, a backstepping method, and low-frequency learning techniques. An NN with an event-triggered controller was developed in [1], with SMC in [13], with backstepping and SMC in [14], and with event-triggered control, backstepping, and prescribed performance control in [15]. In [16] a data-driven USV motion control method based on deep learning with reinforcement was reported. An approach consisting of the radial basis function (RBF) and model reference adaptive control (MEAC) was developed in [17] (verified by experiment). A control strategy using a prescribed time disturbance observer and an auxiliary dynamic system to achieve prescribed time convergence in the presence of external disturbances and input saturation was presented in [18]. Some approaches are based on fuzzy logic as described in [19] (plus event-triggered control (ETC), dynamic surface control), ref. [20] (controller with fuzzy observer). A method using a combination of improved line-of-sight, an improved extended disturbance observer and fuzzy control was developed in [21]. There are also many other control schemes that are composed of different approaches, for example, combining high-gain extended state observer, kinematic and dynamic law [22], the virtual vehicle method, kinematic controllers, and proportional-integral-derivative sliding mode control (PID-SMC) [23]. Three control schemes, which are modified dynamic inversion methods, were introduced in [24]. An approach involving a combination of a fixed-time extended state observer (FESO), a fixed-time differentiator, and a fixed-time sliding mode control (FTSMC) algorithm was proposed in [25]. Prescribed performance guarantee algorithms were presented, e.g., in [26]. An approach to the trajectory tracking problem based on contraction theory was given in [27]. A new continuous desired course angle was designed in [28], which was found to be better than the traditional desired angle design based on tracking errors and excludes the problem of discontinuous desired angles in the traditional design. In addition, based on the desired course angle design, dynamic surface control was introduced, which reduces the computational complexity and allows the desired speed to be obtained. Experimental confirmation of the performance of the saturated adaptive single-parameter backstepping method can be found, e.g., in [29]. It is worth noting, however, that verification of control schemes by means of experiment is rare because simulation approaches dominate in this field.

In studies where the vehicle dynamics model is described by a diagonal inertia matrix, different approaches to the control problem are used. First, various control methods are used and very often combinations of these. Secondly, control algorithms also use components that should provide robustness to changes in model parameters or external disturbances. Their presence should result in the controller's work being correct, i.e., ensuring that the desired trajectory is tracked, despite the aforementioned effects. The developed methods are oriented towards obtaining better performance compared to existing control strategies but assume full symmetry of the model. However,the difficulty of obtaining a fully symmetric model is a serious limitation. This is because it is assumed that the controller will operate properly for the operating conditions that have been included in the algorithm. Unfortunately, there may be a situation in which both the component responsible for parameter uncertainty and for compensating for the effect of environmental inputs will turn out to be insufficient to realize the main purpose of the control, namely, trajectory tracking. But additional motion disturbances can be caused by shifting the center

of mass because there will then be changes in the parameters of the model but also an increase in the effects of the external environment. The cause of these disturbances may be the displacement of a unit of vehicle equipment or other event, such as omission in the model of a certain mass that should be taken into account.

The development of control algorithms involves the use of increasingly sophisticated strategies to increase the performance of the control system. However, it is worth questioning whether such algorithms will also be effective when full vehicle symmetry is not guaranteed. Thus, there is a certain problem of controller robustness to inaccurately known vehicle dynamic parameters. It is also worth noting that the verification of control schemes for fully symmetric vehicle models ends with the use of simulation results. There is then no answer to the question of whether the controller works effectively when the center of mass shifts relative to the geometric center. Works in which the problems arising from the application of a model with a diagonal inertia matrix by means of real experiment are exceptional, e.g., [30,31]. However, it is then possible to evaluate the significance of the control scheme in practice (in the cited work, the method of model predictive control was used).

One solution to ensure that the possibility of shifting the center of mass is taken into account is to assume partial asymmetry of the vehicle model (symmetry in the longitudinal axis). For such a formulated task, there are many solutions, only some of which will be referred to here. Control strategy designs confirmed by simulation tests can be found, for example, in [32–37]. There are studies in which control methods have been tested experimentally. In [38], a linearization method was applied using the hand position point as input, and, based on this, a trajectory tracking strategy was developed. However, it was assumed that the ocean current in the inertial system was constant and non-rotational, which limits the applicability of this strategy. An approach combining NN and the error transformation function to guarantee transient tracking performance was described in [39]. In this case, the disturbance was variable. An adaptive sliding mode control was verified experimentally in [40], whereas an adaptive integral terminal super-twisting algorithm was designed for guidance and control according to a cascade strategy in [41]. In both papers, it was assumed that the off-diagonal terms of the inertial matrices were small compared to the main diagonal terms and could, therefore, be neglected. The method proposed in [42] was based on a model transformation that allowed the derivation of control models for the radius-track, yaw-track and sway-track in a newly defined polar coordinate system moving along a path (PMPCS). If the control method has been experimentally verified, it can be concluded that it guarantees satisfactory performance of the controller (with the introduced limitations and assumptions) and its usefulness is obvious. Also, when the model is partially asymmetric, additional components are often introduced into the controller to ensure robustness to changes in model parameters and environmental factors. But when the verification involves simulation only, then there is no such guarantee. This paper refers to works utilizing simulation validation.

The article proposes a particular scheme for verifying the algorithm with a dynamics model that considers only the diagonal inertia matrix. The control schemes were chosen arbitrarily to verify the proposed procedure; nevertheless, the proposed approach can also be applied to other control algorithms. It is also possible to verify strategies designed for partially asymmetric models; however, this is not the subject of this work, although it is possible. This is a method based on simulations using the control method for fully asymmetric models (with respect to the x-axis and y-axis) [43] and which is useful when a real experiment cannot be carried out. An additional component of the proposed approach is the estimation of the dynamic response with assumed control signals (thrust and torque). Because it is extremely difficult to select appropriate control strategies from the very many existing solutions, we are limited here to using only a certain class of algorithms. The method for testing the effectiveness of trajectory tracking can also be used to study other control schemes. This paper is an attempt to answer the question of whether the control schemes selected for testing a marine vehicle in horizontal motion and designed

for a fully symmetrical model guarantee sufficient performance when the center of mass is shifted.

The contributions of this work include the following:

(1) Proposal of an approach for testing the control efficiency of different algorithms by comparing them based on established criteria.
(2) Verification of the effectiveness of control algorithms for vehicles described by a diagonal inertia matrix based on the technical data of a real marine vehicle, also taking into account limitations of the operating conditions (drives, speeds). For the tested vehicle, realistic operating conditions derived from the literature (e.g., values of driving forces, speeds) were adopted in order to ensure that the results corresponded to real-world situations.
(3) Simulation studies were carried out on a model of a real vehicle, taking into account realizable values of force and torque, as well as velocity, which is not always fulfilled in works concerning the design of control algorithms designed for trajectory tracking.

It is worth noting that the conditions formulated in (2) and (3) make it difficult for the controller to operate because the values of force and torque cannot be arbitrarily increased but are limited to values that can actually be obtained to control the marine vehicle under test. The advantage of this approach is that the results obtained can be close to those possible under real conditions. At the same time, an answer is obtained to the question of what the effectiveness of the tested control algorithm will be in such a situation.

The remainder of this article is organized as follows: In Section 2, the mathematical model of the vehicle under consideration is discussed. The method for analyzing the control algorithms is presented in Section 3. Numerical simulations illustrating the application of the proposed method are described in Section 4. Finally, the conclusions of this work are provided in Section 5.

## 2. Mathematical Model of Marine Vehicle Moving Horizontally

An underactuated marine vehicle sketch in the horizontal plane is presented in Figure 1.

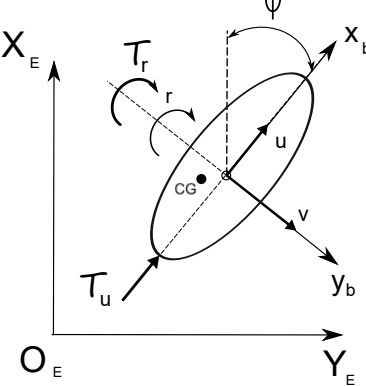

**Figure 1.** Underwater vehicle model sketch.

The mathematical nonlinear model of the vehicle under consideration can be represented as follows [44]:

$$\dot{\eta} = J(\psi)\nu, \tag{1}$$

$$M\dot{\nu} + C(\nu)\nu + D(\nu)\nu = \tau + f_{exd}, \tag{2}$$

where $\eta = [x, y, \psi]^T$ denotes the position in the Earth-fixed frame, $\nu = [u, v, r]^T$ denotes the velocity vector (in surge, sway, and yaw directions in the body-fixed frame), $J(\psi)$ means the transformation matrix, $M$ means the vehicle inertia matrix, $C(\nu)$ means the Coriolis and centripetal matrix, and $D(\nu)$ represents the matrix containing hydrodynamic damping. The control vector $\tau = [\tau_u, 0, \tau_r]^T$ is the thruster forces vector, where $\tau_u$ is the surge force, whereas $\tau_r$ is the yaw torque. The external disturbances are collected in the vector $f_{exd}$.

Elements of the matrices in Equations (1) and (2) are given in Appendix A.

The main disadvantage of (2) is that the matrix $M$ is symmetric in the general case. This form of the matrix causes the inertial forces to be coupled and, therefore, the force and torque influence not only the corresponding acceleration, but also affect other variables that are coupled through the $M$ matrix. An appropriate method would be to diagonalize the $M$ matrix so as to guarantee decoupling of inertial forces to facilitate the work of the controller. One method of diagonalization is to express the equations of motion in inertial quasi-velocities (IQV) which leads to diagonal equations in which decoupling is obtained in the sense of quasi-forces.

**IQV-based equations and equations with diagonal inertia matrix**. In order to keep the symmetry of the $M$ matrix, it is assumed that all model inaccuracies $f_m$ and external disturbances $f_{exd}$ are included in the function $f_{te} = f_m + f_{exd}$. Then, decomposition of the symmetric matrix $M$ leads to a diagonal matrix $\mathbb{A} = \hat{\mathbb{P}}^T M \hat{\mathbb{P}}$ [45,46]. Moreover, $M = \hat{\mathbb{P}}^{-T} \mathbb{A} \hat{\mathbb{P}}$. The $\hat{\mathbb{P}}$ matrix contains nominal parameters, while any inaccuracies of $\hat{\mathbb{P}}$ are shifted to the vector $f = f_{te} + \Delta\hat{\mathbb{P}}$ defined as $f = [f_u, f_v, f_r]^T$. On the other hand, the decomposition of the matrix $\hat{M}$ with nominal parameters yields a matrix $\hat{\mathbb{N}} = \hat{\mathbb{P}}^T \hat{M} \hat{\mathbb{P}}$.

The new equations instead of (2) are of the form (cf. [43]):

$$\mathbb{A}\dot{\zeta} + \hat{\mathbb{P}}^T C(v)v + \hat{\mathbb{P}}^T D(v)v = \hat{\mathbb{P}}^T \tau + \hat{\mathbb{P}}^T f, \tag{3}$$

$$v = \hat{\mathbb{P}}\zeta, \tag{4}$$

$$\hat{\mathbb{P}} = \begin{bmatrix} 1 & 0 & \hat{\mathbb{P}}_{13} \\ 0 & 1 & \hat{\mathbb{P}}_{23} \\ 0 & 0 & 1 \end{bmatrix}, \quad \mathbb{A} = \text{diag}\{\mathbb{A}_1, \mathbb{A}_2, \mathbb{A}_3\}, \tag{5}$$

$$\mathbb{A}_1 = m_{11}, \quad \mathbb{A}_2 = m_{22}, \quad \mathbb{A}_3 = m_{33} - (m_{13}^2/m_{11}) - (m_{23}^2/m_{22}), \tag{6}$$

$$\hat{\mathbb{P}}_{13} = -(\hat{m}_{13}/\hat{m}_{11}), \quad \hat{\mathbb{P}}_{23} = -(\hat{m}_{23}/\hat{m}_{22}). \tag{7}$$

The vector of the inertial quasi-velocities (IQV) is defined as $\zeta = [\zeta_1, \zeta_2, \zeta_3]^T$. Equations (3) and (4) can also be expressed in the form shown in Appendix A.

**Equations of motion with diagonal inertia matrix**. A common method of simplifying the model is to ignore couplings, resulting in a diagonal inertia matrix. Controller design is less complicated, but the disadvantage of this approach is that the controller may not work effectively when the couplings are strong enough that they should be included in the vehicle model and controller equations. The equation describing the vehicle dynamics with a diagonal inertia matrix, i.e., reduced Equation (2), has the form:

$$M_D \dot{v} + C_D(v)v + D_D(v)v = \tau + f_{exd}, \tag{8}$$

in which:

$$M_D = \begin{bmatrix} m_{11} & 0 & 0 \\ 0 & m_{22} & 0 \\ 0 & 0 & m_{33} \end{bmatrix}, \quad C_D(v) = \begin{bmatrix} 0 & 0 & c_{D13} \\ 0 & 0 & c_{D23} \\ -c_{D13} & -c_{D23} & 0 \end{bmatrix},$$

$$D_D(v) = \begin{bmatrix} d_{11} & 0 & 0 \\ 0 & d_{22} & 0 \\ 0 & 0 & d_{33} \end{bmatrix}, \tag{9}$$

where $c_{D13} = -m_{22}v$ and $c_{D23} = m_{11}u$. The corresponding equations but not expressed in matrix-vector form are as follows:

$$m_{11}\dot{u} = m_{22}vr - d_{11}u + \tau_u + f_u, \tag{10}$$

$$m_{22}\dot{v} = -m_{11}ur - d_{22}v + f_v, \tag{11}$$

$$m_{33}\dot{r} = (m_{11} - m_{22})uv - d_{33}r + \tau_r + f_r. \tag{12}$$

**Remark 1.** *Because of the description of the dynamics for two different systems (with a symmetric and a diagonal matrix), the elements of vectors v and v̇ will be different in Equations (A6)–(A8), and (10)–(12). It is this difference that provides the basis for further analysis of vehicle dynamics.*

### 3. Control Analysis Method Based on Dynamics Model

This section discusses a method of analyzing the equations of motion to test dynamic models and the corresponding control schemes to test the suitability of these algorithms for moving the center of mass of a vehicle.

The proposed method for evaluating the suitability of control strategies consists of several steps:

1.  Examining the suitability of the controller along with the vehicle dynamics model by means of a test.
2.  Brief analysis and discussion of the equations of motion.
3.  Determining assumptions and selecting evaluation criteria.

The first point addresses the following issue. The usefulness of the controller together with an appropriate dynamic model of the vehicle is a key issue because it allows to answer the question whether a control scheme known from the literature can guarantee acceptable performance if its operating conditions are changed (e.g., a different vehicle model, a different desired trajectory). If the answer is positive, then such an algorithm, due to a certain universality, can be used in a wider range than proposed in the source work. In order to make the comparison possible, a benchmark algorithm is applied, which was originally designed [43] for a vehicle model with asymmetry. For consideration of this issue, the procedure described in Section 3.1 is proposed.

The second point concerns the following issue. Based on the analysis, differences can be seen between the equations of motion for the model with a symmetric inertia matrix, in which shifts of the center of mass were taken into account, and a diagonal inertia matrix, in which these displacements were ignored. Thus, information is also available about how the dynamics of the system will be modified due to the lack of symmetry of the vehicle model. This problem is considered in Section 3.2.

The third point is related to the assumptions of the performance evaluation criteria. The selected simulation tests will generate results for each control scheme. In order to obtain a basis for their evaluation, the differences arising from the model for an asymmetric vehicle and a fully symmetric one should be indicated. Although this is not a primary index, it is worth paying attention to. On the other hand, the ground for comparing the performance of different control schemes is the time history of selected variables considered relevant to control and the selection of criteria for evaluating this performance. This issue was developed in Section 3.3.

The mentioned stages will be discussed in the following subsections.

*3.1. Controller Usability Test Using Vehicle Dynamics Model*

A scheme of the procedure for evaluating the usefulness of control algorithms for asymmetric vehicles in horizontal motion proposed in this article is shown in Figure 2. This method is performed in several stages:

1.  At the first stage, a vehicle is selected whose equations of motion include a symmetric inertia matrix but also, in reduced form, a diagonal matrix. Two such models are considered. The test conditions are also established as follows: maximum thrust, trajectory, initial conditions, and other operating conditions of the controller. A benchmark control scheme is selected that is suitable for a fully asymmetric vehicle (in the $x$ and $y$ directions). Control strategies designed for a diagonal vehicle model are also selected.
2.  At the second stage, the trajectory of the vehicle's movement is generated according to the assumed operating conditions.
3.  At the third stage, the performance of the control algorithm selected as a benchmark is tested. The resulting output signals in the form of force and torque and selected

quantities representing the performance of the algorithm are archived for the tasks carried out in the subsequent stages of the procedure.

4.  At the fourth stage, the force and torque signals are estimated into a mathematical form that replaces their real time history. On this basis, the effect of control signals on the formation of disturbances due to inertial couplings, which are not taken into account in models with a diagonal inertia matrix, is analyzed.

5.  At the fifth stage, the chosen control scheme is tested for its effectiveness when inertial couplings are present in the dynamics model. This means that the controller considers a model with a diagonal inertia matrix, while the vehicle dynamics are described by a model with a symmetric inertia matrix.

6.  In the final stage, the performance of the reference controller is compared with that of a controller designed for a fully symmetrical vehicle model. In this process, predetermined criteria are used. The result of this procedure is an answer as to whether the tested algorithm is effective when the full symmetry of the model cannot be guaranteed.

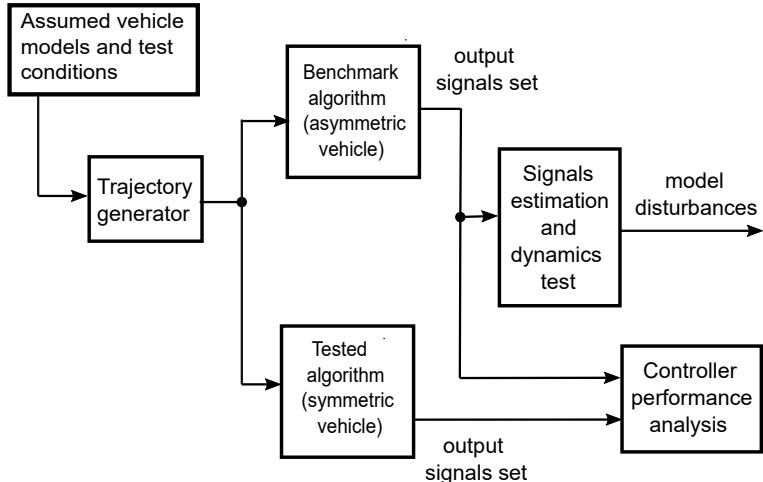

**Figure 2.** Research procedure scheme. The term "set of output signals" means the set of output signals relevant to the test and obtained using the control algorithm under test.

Figure 3 shows the part of the method that relates to the comparative analysis of the symmetric dynamics model and the diagonal model, i.e., 'signals estimation and dynamics test'. This stage is carried out as follows:

1.  Formulating the purpose of the test: In this part of the method, the performance of the control algorithms is not compared, but the difference in responses obtained from the symmetric and diagonal model (due to the inertia matrix) is studied.

2.  The *Fu* and *Fr* signals refer to the estimated force in the longitudinal direction and torque, respectively. These quantities are defined after obtaining $\tau_u$ and $\tau_r$ signals from the controller designed for the asymmetric vehicle model. Based on the time history of $\tau_u$ and $\tau_r$, an approximation of these functions is performed. The *Fu* and *Fr* functions are expressed by a constant value (average value of force or torque, i.e., mean value and variable component depending on standard deviation) in the following form:

$$Fu = \text{mean}(\tau_u) + (A_1 \sin(a_{u1}t) + A_2 \cos(a_{u2}t)) \cdot \text{std}(\tau_u), \quad (13)$$

$$Fr = \text{mean}(\tau_r) + (A_3 \sin(a_{r1}t) + A_4 \cos(a_{r2}t)) \cdot \text{std}(\tau_r), \quad (14)$$

where $a_{u1}, a_{u2}, a_{r1}, a_{r2}, A_1, A_2, A_3, A_3$ denote some constants which are chosen arbitrarily so as to achieve likeness to the original function (not the same function but just similar) to maintain visual similarity to the force and torque. For the analysis, only the estimation of $\tau_u$ and $\tau_r$ is needed to determine what disturbances result from

the presence of off-diagonal components and which the algorithm must overcome to achieve acceptable results.

3. Comparison of signals representing inertial forces for symmetric and diagonal matrix and determination of differences in their values, namely: $\delta_1 = m_{11}(\dot{u} - \dot{u}_{diag})$, $\delta_2 = m_{22}(\dot{v} - \dot{v}_{diag})$, $\delta_{3T} = m_{33}(\dot{r} - \dot{r}_{diag})$.

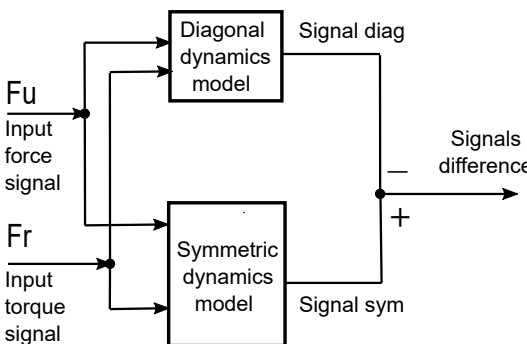

**Figure 3.** Diagram of vehicle dynamics test based on output signals estimation chart. "Signal diag"—the set of output signals obtained from a model with a diagonal inertia matrix, while "Signal sym"—the set of signals obtained for a model with a symmetric inertia matrix.

On the basis of the results obtained representing the differences between the inertial forces in the model with a symmetric inertia matrix and a diagonal matrix information is obtained about additional forces that are disturbances and result from dynamic couplings. Such forces must reduce the control algorithm when it is designed for a vehicle with full symmetry.

### 3.2. Brief Analysis and Discussion of Equations of Motion

If the geometric center of the vehicle is not identical to the center of mass, the resulting errors must be included in the dynamic equations. In equations expressed in IQV, such a situation is taken into account. On the other hand, the assumption that the inertia matrix $M$ is diagonal is a simplification that can cause a control algorithm based on such a model to fail. Therefore, the control scheme must compensate for errors resulting from the simplification of the dynamics model. Comparing Equations (A6)–(A8) with (10)–(12), it can be written:

$$m_{11}\dot{u} = m_{22}vr - d_{11}u + \tau_u + f_u + \sigma_1, \tag{15}$$

$$m_{22}\dot{v} = -m_{11}ur - d_{22}v + f_v + \sigma_2, \tag{16}$$

$$m_{33}\dot{r} = (m_{11} - m_{22})uv - d_{33}r + \tau_r + f_r + \sigma_3 + \Delta\tau_u + \Delta f_{uv}, \tag{17}$$

where:

$$\sigma_1 = m_{23}r^2 - d_{13}r + m_{11}\hat{\mathbb{P}}_{13}\dot{r}, \tag{18}$$

$$\sigma_2 = m_{13}r^2 - d_{23}r + m_{22}\hat{\mathbb{P}}_{23}\dot{r}, \tag{19}$$

$$\sigma_3 = -(m_{13}v + m_{23}u)r + \hat{\mathbb{P}}_{13}(m_{22}v + m_{23}r)r - \hat{\mathbb{P}}_{23}(m_{11}u - m_{13}r)r - (\hat{\mathbb{P}}_{13}d_{11} + d_{31})u$$
$$- (\hat{\mathbb{P}}_{23}d_{22} + d_{32})v - (\hat{\mathbb{P}}_{13}d_{13} + \hat{\mathbb{P}}_{23}d_{23})r + \left((m_{13}^2/m_{11}) + (m_{23}^2/m_{22})\right)\dot{r}, \tag{20}$$

$$\Delta\tau_u = \hat{\mathbb{P}}_{13}\tau_u, \quad \Delta f_{uv} = \hat{\mathbb{P}}_{13}f_u + \hat{\mathbb{P}}_{23}f_v, \tag{21}$$

$$\delta_{3T} = \delta_3 + \Delta\tau_u + \Delta f_{uv}. \tag{22}$$

Based on the above equations, the following observations can be made:

1. If the center of mass will not be the geometric center at the same time, then it is necessary to add a compensating component. This means that the control algorithm:

   (a) should also include a component to offset model inaccuracies, or

(b)   be designed to effectively perform the control task despite modeling inaccuracies.

2.   Disturbance components $\delta_1$ and $\delta_2$ depend on the velocity $r$ and its time derivative. From this, it can be concluded that as long as this velocity has very small values, e.g., 0.1 rad/s or slightly higher, the values of these components will not disable the control of the vehicle. Thus, if the algorithm is sufficiently robust to disturbances, trajectory tracking in the $x$ and $y$ directions will be effective. If the necessary condition of the algorithm is a diagonal inertia matrix without the possibility of correcting model inaccuracies, then such an algorithm will fail.

3.   The third Equation (17) is not so easy to analyze.

(a)   The perturbation component $\delta_3$ depends on the entire velocity vector and, therefore, its role is not easy to determine. Instead, it can be deduced that its values will depend on the distance of the center of mass from the geometric center of the vehicle. This means that with a small difference in this distance, the values of $\delta_3$ will be able to be compensated for with a fault-tolerant control scheme. Otherwise, this will be impossible or much more difficult. When the difference in the distance of the two centers increases to a critical value, the algorithm will fail. Estimating this critical value is possible with a large amount of calculation, and it is not known whether such a value will allow correct control of the vehicle.

(b)   Unfortunately, the second problem arising from the equation is the presence of $\Delta\tau_u$ and $\Delta f_{uv}$ disturbances. It can be seen that the rotational motion is influenced by both the driving force in the forward direction and the inaccuracies of the model. The disturbances $f_u$ and $f_v$ also change the rotational motion and these changes are closely related to the inaccuracy of the model. Based on knowledge of the values of these components, it is not possible to answer conclusively whether they will disable the controller. Their presence, however, indicates the need to include such disturbances in the equation of the algorithm.

In conclusion, it can be said that the control of speed $r$ is more difficult than linear speeds, and only a properly constructed algorithm will allow effective control. It may also happen that linear positions will be tracked correctly (the convergence of tracking errors will be observed) but the rotation velocity will not be tracked.

*3.3. Evaluation Criteria*

The criteria can be divided into those relating to the dynamics model and evaluation of the control scheme.

3.3.1. Estimation of Model Disturbance Dynamics

In order to estimate the differences in the dynamics of a system with a symmetric inertia matrix and a diagonal matrix, the following procedure is proposed. Equations (A6)–(A8) are rewritten to the form:

$$m_{11}\dot{u} = H_1(v) + \tau_u + f_u + \hat{\mathbb{P}}_{13}\dot{r}, \tag{23}$$

$$m_{22}\dot{v} = H_2(v) + f_v + \hat{\mathbb{P}}_{23}\dot{r}, \tag{24}$$

$$m_{33}\dot{r} = H_3(v) + \tau_{\zeta_3} + f_{\zeta_3} + \left( (m_{13}^2/m_{11}) + (m_{23}^2/m_{22}) \right)\dot{r}. \tag{25}$$

Denoting the right-hand sides of Equations (23)–(25) as $\Psi_1$, $\Psi_2$, $\Psi_3$ and Equations (10)–(12) as $\Psi_{D1}$, $\Psi_{D2}$, $\Psi_{D3}$, respectively, the differences between them can be determined as:

$$\delta_1 = \Psi_1 - \Psi_{D1}, \quad \delta_2 = \Psi_2 - \Psi_{D2}, \quad \delta_{3T} = \Psi_3 - \Psi_{D3}. \tag{26}$$

Estimation of these components is performed by applying maximum values of force and torque to the input of the system. Calculation of the function (26) is necessary to compare with the propulsion capabilities of the vehicle (force and torque). Based on the values obtained, it is possible to assess what disturbances resulting from the asymmetry of the model the control algorithm must overcome.

3.3.2. Estimation of Control Scheme Performance

The following set of metrics was selected to assess the controller's performance:

(1)  time history of selected variables and tracking errors;
(2)  mean of errors and their standard deviation mean (a), std (a), where $a = x_e, y_e$;
(3)  mean integrated absolute error defined as $MIA = \frac{1}{t_f - t_0} \int_{t_0}^{t_f} |a(t)| dt$, where $a = x_e, y_e$;
(4)  mean integrated absolute control defined as $MIAC = \frac{1}{t_f - t_0} \int_{t_0}^{t_f} |\tau(t)| dt$;
(5)  root mean square of the tracking error defined as $RMS = \sqrt{\frac{1}{t_f - t_0} \int_{t_0}^{t_f} \|e(t)\|^2 dt}$, where $\|e(t)\| = \sqrt{(x_e)^2 + (y_e)^2}$ ($x_e, y_e$ are the position errors in the body frame);
(6)  mean kinetic energy defined as mean $K$.

The time history of the selected variables is necessary to determine whether the control algorithm is working properly. Other indexes are used to compare the performance of various control schemes.

## 4. Simulation Results

The purpose of the proposed tests is to demonstrate the performance of selected control schemes and the scheme assumed as the basic one (i.e., developed for a model with a shifted center of mass). For this task, the vehicle model used in marine research was taken as the real one. However, dimensions representing the shifted center of mass were added to determine whether the algorithms would work correctly in the new situation. Limitations due to the thrusters used in the vehicle were also considered. In addition, it was assumed that two types of trajectories would be tracked, namely, linear and nonlinear. The choice of two trajectories avoids biased results that could occur if only one trajectory is tracked. The influence of environmental factors was also accounted for by means of disturbance functions.

In order to determine the effectiveness of the selected control algorithms, graphs were taken of the desired and realized trajectory as well as of position errors. In addition, vehicle velocity, force and torque were taken into account to determine whether they had values greater than acceptable, and kinetic energy values to evaluate its consumption. For the basic scheme, quasi-velocity errors were added, which makes it possible to estimate the deformation of velocity during vehicle movement. The tests were completed by evaluating the performance of the controllers using the assumed criteria.

### 4.1. Vehicle Model and Test Conditions

An underwater vehicle called C-Ranger AUV was selected for testing. This is a vehicle used for underwater research [47], so it can be assumed that the parameters apply to the real system. The vehicle parameters given in Table 1 can be found in [48,49].

In order to consider the matrix $M$ with off-diagonal elements, it is assumed that $m_{13} = m_{31} = 13$ kgm and $m_{23} = m_{32} = 25$ kgm (in the cited references, these values are not taken into account but they are needed for this test). These data correspond to the values of $x_g = 0.08$ m, $y_g = -0.02$ m, and $Y_{\dot{r}} = 8.5$ kgm, $X_{\dot{r}} = 8.9$ kgm. Dynamic couplings were estimated at less than 5.5%, which means they are slightly more than very weak. This parameters set allows to calculate elements of the diagonal matrix $N$, i.e., $N_1 = 273.8$ kg, and $N_2 = 273.8$ kg, $N_3 = 25.7$ kgm$^2$. The other non-zero linear and quadratic coefficients were assumed as: $X_r = 10, X_{|u|r} = 5, Y_r = 10, Y_{|v|r} = 5, N_u = 5, N_{|r|u} = 0.5, N_v = 5, N_{|r|v} = 0.5$.

**Table 1.** Parameter values of C-Ranger used in the tests [48,49].

| Parameter | Value | Parameter | Value |
|-----------|-------|-----------|-------|
| $L$ | 1.6 m | $X_u$ | 120 Ns/m |
| $b$ | 1.3 m | $Y_v$ | 90 Ns/m |
| $h$ | 1.1 m | $N_r$ | 18 Nms |
| $m$ | 206 kg | $X_{|u|u}$ | 90 Ns$^2$/m$^2$ |
| $m_{11}$ | 273.8 kg | $Y_{|v|v}$ | 90 Ns$^2$/m$^2$ |
| $m_{22}$ | 273.8 kg | $N_{|r|r}$ | 15 Ns$^2$/m$^2$ |
| $m_{33}$ | 28.6 Nms$^2$ | | |

The force and torque limits as well as the forward velocity are applied based on vehicle specifications [48,49]. The forward force is $\tau_u = T_1 + T_2$ (maximum values of forces from engines $T_1 = T_2 = 100$ N) and torques $\tau_r = T_1 l_2 - T_2 l_2$, where $l_2 = 0.410$ m). The maximum values of the propulsion force and torque were assumed as follows: $|\tau_u| \leq 120$ N and $|\tau_r| \leq 32$ Nm. The assumed maximum value of longitudinal speed is $u_{max} = 1.5$ m/s (close to 3 knots). By continuous running time, the vehicle can travel for eight hours at a speed of 1 knot.

The simulations using Matlab/Simulink were performed for time $t = 120$ s (the time step $\Delta t = 0.05$ s, and using the method ODE3 Bogacki–Shampine). The following desired trajectories were tested: linear and complex:

$$p_{d1} = [0.5\,t,\ \ 0.1\,t]^T, \tag{27}$$

$$p_{d2} = [0.4\,t,\ \ 10\sin(0.03\,t) + 5\cos(0.02\,t) + 0.1\,t]^T, \tag{28}$$

with initial points $p_{0d1} = [0.5,\ 2]^T$ ($\psi_{0d1} = 0$), $p_{0d2} = [-1,\ 8]^T$ ($\psi_{0d2} = 0$), respectively.

The disturbance functions considered for both vehicles were of the form:

$$\begin{bmatrix} f_u(t) \\ f_v(t) \\ f_r(t) \end{bmatrix} = \begin{bmatrix} 15 + 5\sin(0.03\,t)\ \text{N,} \\ -10\sin(0.01\,t)\ \text{N,} \\ 5\sin(0.02\,t) + 0.5\cos(0.05\,t)\ \text{Nm} \end{bmatrix} \tag{29}$$

*Assumption for simulation.* If the thrust saturation effect is not serious then it can be taken that $\tau_u$ and $\tau_r$ are bounded, i.e., $\tau_{u\,min} \leq \tau_u \leq \tau_{u\,max}$ and $\tau_{r\,min} \leq \tau_r \leq \tau_{r\,max}$. Such assumptions concerning control signals boundedness are made for marine vehicles, for example in [50,51]. This is allowed when the control input values exceed the thruster limit only sometimes [52]. In the simulation tests, the operating conditions, as well as the desired trajectories, were chosen to meet this condition.

*Comment.* Controller gains can be selected using the heuristic method explained in [53], which allows tuning the controller parameters to improve its quality as measured by the time history of the selected signals and the assumed evaluation criteria. The advantage of this method is that it is suitable for different types of control schemes (including those designed for underactuated vehicle models), and is, therefore, useful when it is necessary to compare the performance of different controllers. This method was applied to simulation testing.

### 4.2. Analysis of Vehicle Model Dynamics Using a Basic Controller

The algorithm proposed in [43] was selected as a benchmark control scheme. It is designed for the control of asymmetric vehicles and proved effective in simulation studies. It employs a model of dynamics described by Equations (3)–(A11) and is based on software from [54]. The following set of controller parameters was selected:

$$\text{Linear trajectory}\ k_1 = 1.0,\ k_2 = 0.1,\ k_3 = 30,\ k_4 = 2.5,\ k_5 = 30; \tag{30}$$

$$\text{Complex trajectory}\ k_1 = 0.8,\ k_2 = 0.15,\ k_3 = 30,\ k_4 = 1.5,\ k_5 = 30. \tag{31}$$

Figure 4a shows that the controller correctly tracks the linear trajectory, and from Figure 4b, it can be seen that the error value of the uncontrolled variable $y$ is close to zero after about 50 s (for the controlled variable $x$, it is about 10 s). The velocity values are acceptable for the test vehicle, as shown in Figure 4c. The force value at the beginning only is maximum and then quickly decreases whereas the torque values are very small (Figure 4d). From Figure 4e, one can note that the kinetic energy is dissipated almost entirely by the forward motion (velocity direction $u$). The quasi-velocity error values from Figure 4f indicate that the velocity deformation is negligible (up to about 0.03 m/s). Hence, it follows that dynamic couplings have very little effect on the movement of the vehicle.

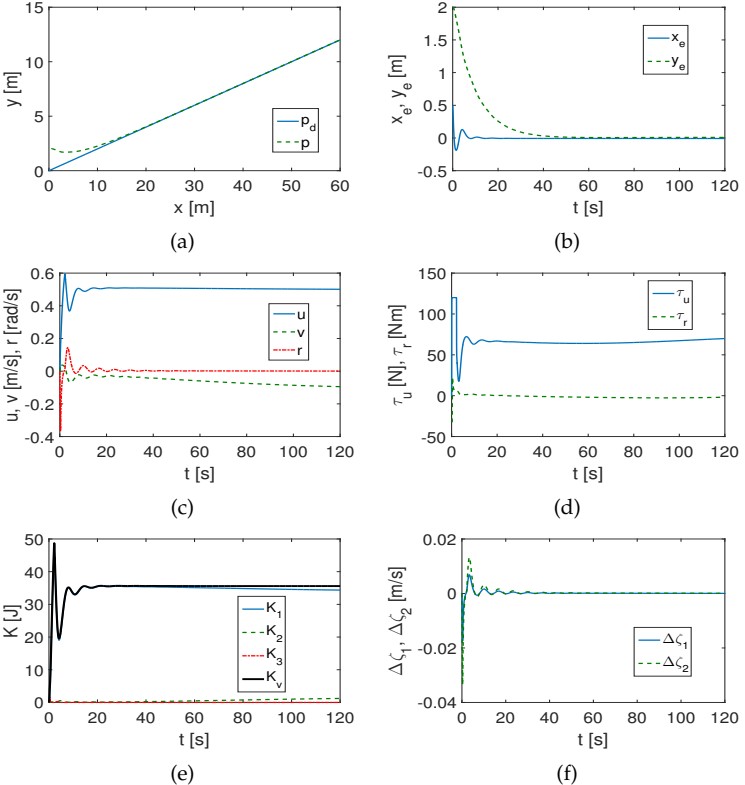

**Figure 4.** Results for QV basic controller and linear trajectory: (**a**) desired and realized trajectory; (**b**) position errors; (**c**) velocities; (**d**) applied force and torque; (**e**) kinetic energy time history; (**f**) quasi-velocity errors $\Delta\zeta_1, \Delta\zeta_2$.

For the complex trajectory, as can be seen from Figure 5a,b, the tracking controller is working properly, but the reduction in the errors of the uncontrollable variable occurs already after about 30 s and, therefore, faster than before. The velocities shown in Figure 5c have acceptable values. The time history of the force and torque are similar to that of a linear trajectory (Figure 5d), but one can see the effect of changing the shape of the trajectory on the movement of the vehicle (waveform). The kinetic energy is dissipated mainly by forward motion (Figure 5d), and Figure 5e shows that dynamic couplings have a negligible effect on vehicle motion.

The evaluated dynamic functions are of the form (for linear trajectory):

$$Fu = 66.2 + 9.0 \cdot 0.3 \sin(-0.03t), \tag{32}$$

$$Fr = -1.3 + 2.5 \cdot 0.4 \sin(-0.02t), \tag{33}$$

and for complex trajectory:

$$Fu = 57.4 + 13.8(\cos(0.06t) + 0.5 \sin(-0.01t)), \tag{34}$$

$$Fr = -1.6 + 2.5 \cdot 0.5 \sin(-0.02t). \tag{35}$$

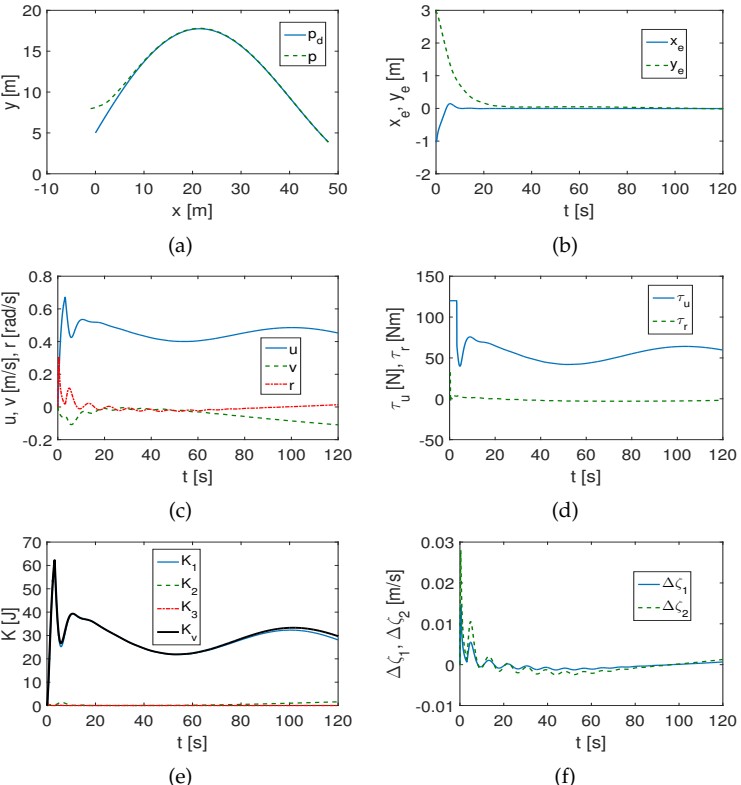

**Figure 5.** Results for QV basic controller and complex trajectory: (**a**) desired and realized trajectory; (**b**) position errors; (**c**) velocities; (**d**) applied force and torque; (**e**) kinetic energy time history; (**f**) quasi-velocity errors $\Delta\zeta_1, \Delta\zeta_2$.

The analysis of the effect of dynamics according to the directions in Section 3.3.1 is shown in Figure 6a,b. It can be clearly seen from them that at the beginning of the movement only the differences in the forces and torque of the model with symmetric and diagonal matrix are small for both linear and complex trajectories. However, it is worth noting that it is the lateral movement that primarily causes the increase in forces (variable $\delta_2$). This direction, therefore, poses the greatest problem for the tracking controller.

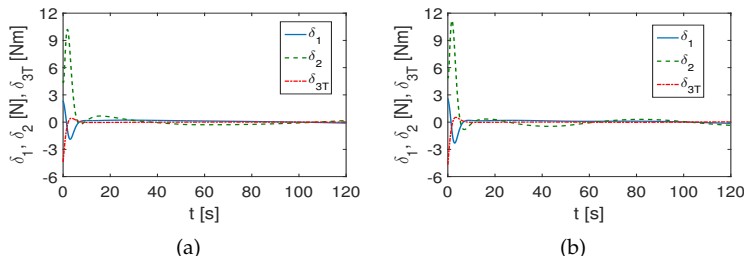

**Figure 6.** Results for dynamic force and torque evaluation: (**a**) linear trajectory; (**b**) complex trajectory.

### 4.3. Tracking Control Algorithms Selected for Comparison

The tracking control schemes selected here are mainly based on the sliding mode control method. Methods of this type often guarantee immunity to disturbances and imprecise known parameters. For this reason, they are suitable for testing. In addition, the cited literature does not indicate exactly how the parameters of the controller were selected, which suggests that it is intuitive or straightforward. However, the primary criterion adopted for comparative testing was the use of a dynamics model with a diagonal inertia matrix. Thus, an answer was sought to the question of the suitability of control schemes suitable for this model when there is a slight asymmetry caused by additional conditions.

Five control schemes were chosen for testing, four of which are modifications of the SMC. Methods of this type are used because of their robustness to disturbances. The basic algorithm is also a modification of SMC, which allows some comparison of algorithms of this type. Nevertheless, the proposed approach can also be used to test control algorithms based on other methods.

### 4.3.1. Adaptive Dynamical Sliding Mode Controller

The strategy from [8] (denoted here as XWQ2015) is a combination of the backstepping technique and SMC. The results shown in the source paper indicate that it is effective and robust to the time-varying disturbances. The dynamics model can be described by Equations (10)–(12). The control scheme was implemented in [55] that includes the relevant software. In the present work, it was adapted to the C-Ranger vehicle. The following set of controller parameters was selected:

$$\text{Linear trajectory } k_1 = k_2 = 0.8, \quad k_3 = 0.1, \quad c_1 = 0.8, \quad c_2 = 1.3, \tag{36}$$
$$k_{s1} = k_{s2} = 0.5, \quad w_{s1} = w_{s2} = 1.0;$$
$$\text{Complex trajectory } k_1 = k_2 = 1.0, \quad k_3 = 0.1, \quad c_1 = 0.8, \quad c_2 = 1.3, \tag{37}$$
$$k_{s1} = k_{s2} = 0.5, \quad w_{s1} = w_{s2} = 1.0.$$

For a linear trajectory, the algorithm works, but there are significant disturbances at the beginning of the movement, as can be seen from Figure 7a,b. Also, the time of error setting is longer at about 60 s. Disturbances resulting from couplings have a significant impact on the velocities, driving forces, and kinetic energies (there are oscillations of these variables), as can be observed in Figure 7c–e.

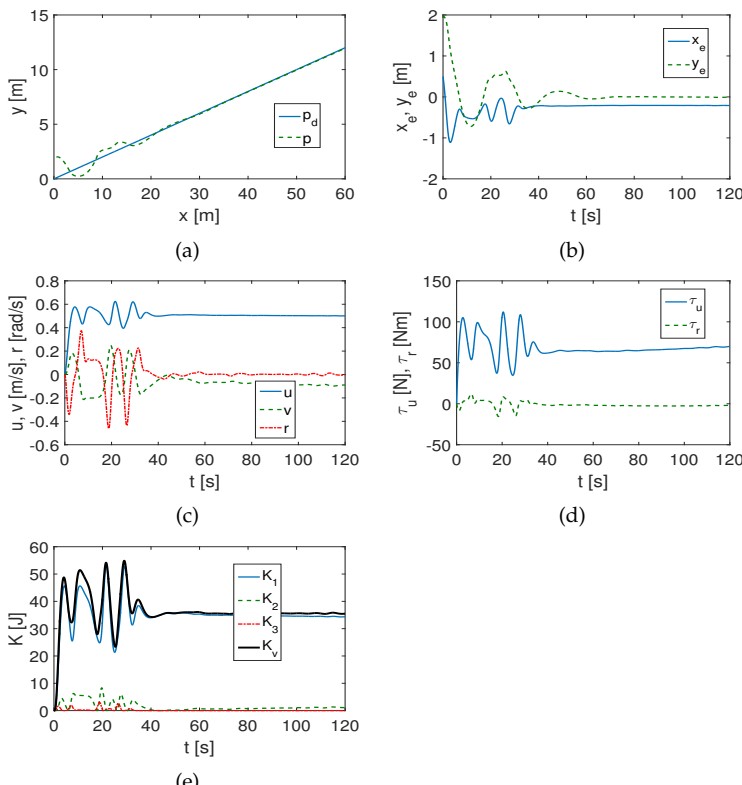

**Figure 7.** Results for XWQ2015 controller and linear trajectory: (**a**) desired and realized trajectory; (**b**) position errors; (**c**) velocities; (**d**) applied force and torque; (**e**) kinetic energy time history.

In the case of a complex trajectory, the algorithm (Figure 8a–e) performs slightly worse with tracking but after about 80 s, it gives results that are acceptable. Previously, the algorithm tended toward steady-state motion, but due to parameter perturbations

and dynamic coupling, there were many oscillations. However, it is worth noting that in steady-state motion, the errors in the lateral direction are large, so that the vehicle does not reach a position close to the target (the errors are noticeably larger than for the base algorithm Figures 7a,b and 8a,b). In addition, for more than 30 s, the value of the force in the longitudinal direction is maximum or only temporarily lower (Figure 8d).

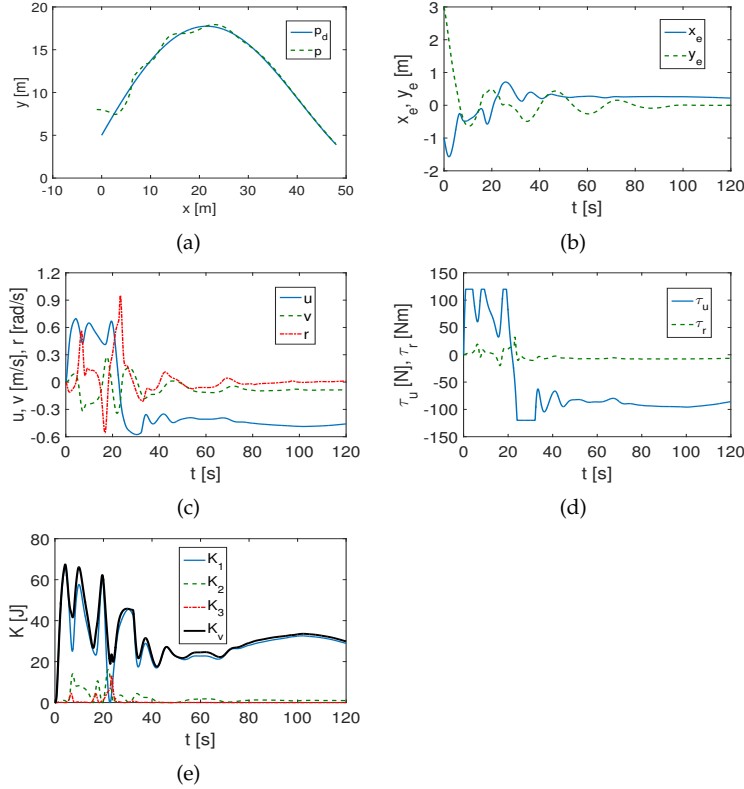

**Figure 8.** Results for XWQ2015 controller and complex trajectory: (**a**) desired and realized trajectory; (**b**) position errors; (**c**) velocities; (**d**) applied force and torque; (**e**) kinetic energy time history.

### 4.3.2. Proportional Integral Sliding Mode Controller with Backstepping

This control scheme, which is based on proportional integral (PI) sliding mode control together with the backstepping technique, comes from [11] (denoted here as SZQZ2018) and was implemented in [54], where the corresponding software was included. Here, it has been customized for the C-Ranger vehicle. The algorithm uses a dynamics model described by Equations (10)–(12). For this control scheme, the following parameters were selected:

$$\text{Linear trajectory } k_1 = 1.0, \ k_2 = 0.1, \ k_3 = 30, \ k_4 = 2.5, \ k_5 = 30; \tag{38}$$

$$\text{Complex trajectory } k_1 = 10, \ k_2 = 10, \ k_3 = 1.0, \ k_4 = 10, \ k_5 = 10. \tag{39}$$

As seen from Figure 9a,b, the linear trajectory is satisfactorily tracked but the velocity, force and torque and kinetic energy changes are oscillatory (Figure 9a,b). This is the result of adjusting the controller's response to couplings in the vehicle model.

Unfortunately, when using the complex trajectory, the algorithm fails completely. As shown in Figure 10a,b, the errors have large values and are non-decreasing. The velocities, force and torque change are oscillatory and with a large value of the amplitude of these changes (Figure 10c–e). Such changes in force and torque values (Figure 10d) would, in the case of a real situation, lead to the destruction of the controller.

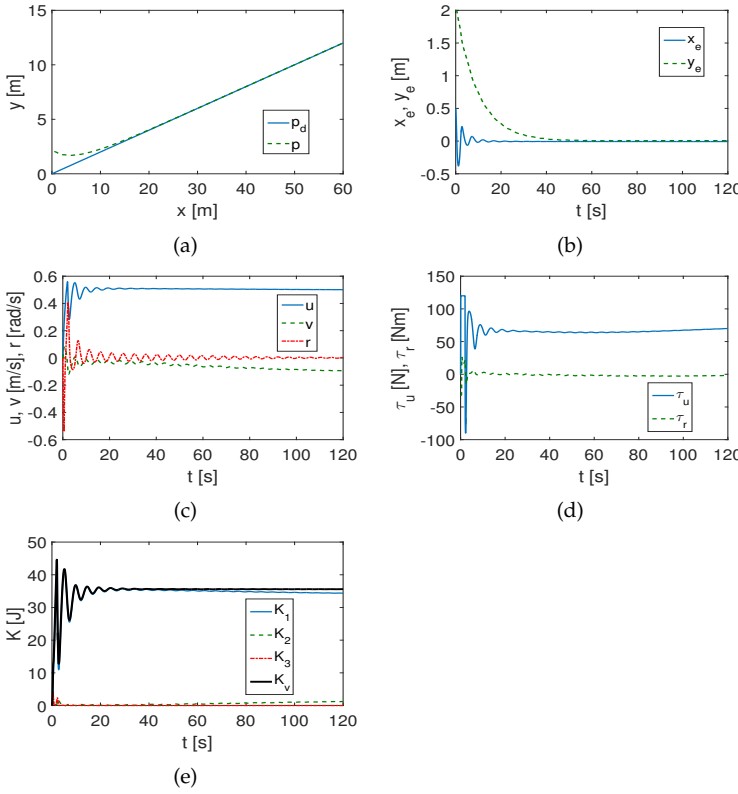

**Figure 9.** Results for SZQZ2018 controller and linear trajectory: (**a**) desired and realized trajectory; (**b**) position errors; (**c**) velocities; (**d**) applied force and torque; (**e**) kinetic energy time history.

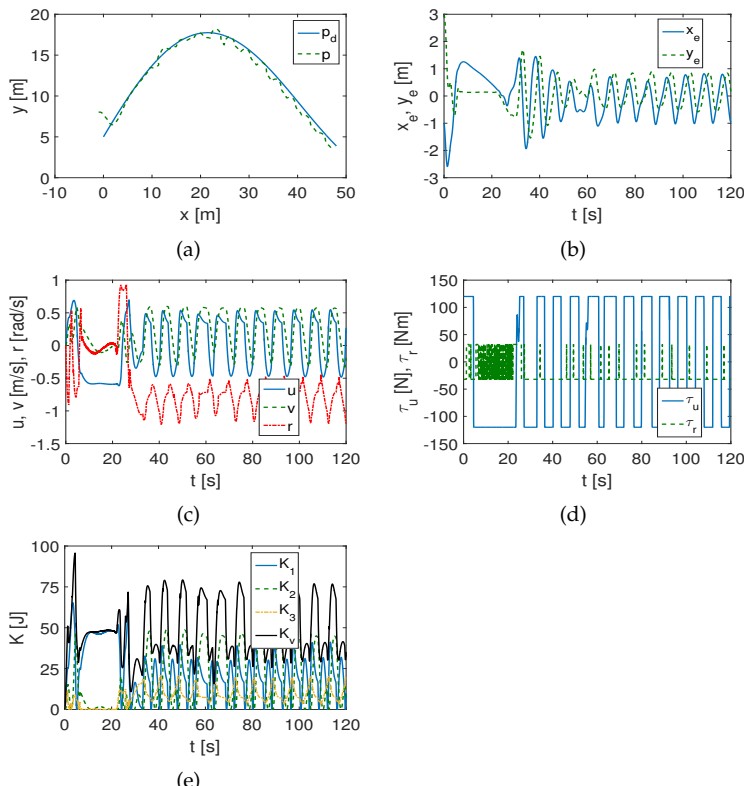

**Figure 10.** Results for SZQZ2018 controller and complex trajectory: (**a**) desired and realized trajectory; (**b**) position errors; (**c**) velocities; (**d**) applied force and torque; (**e**) kinetic energy time history.

### 4.3.3. Sliding Mode Controller Robust against Bounded Disturbances

The controller proposed in [3] (denoted here as EZY2016) is based on sliding mode trajectory tracking control and the equations of motion (10)–(12), but in which the components of the disturbance forces $f_u, f_v, f_r$ are omitted and the quantities $d_{11}, d_{22}, d_{33}$ are limited to constant values only. However, the cited work demonstrates the robustness of the proposed trajectory tracking control scheme to various types of disturbances, with satisfactory results. Software from [56] was used here to perform the comparison tests, but adapted to the C-Ranger vehicle. The following controller parameters were chosen:

$$\text{Linear trajectory } l_x = l_y = 15, \; k_x = k_y = 2.0, \; \lambda_1 = \lambda_2 = \lambda_3 = 12, \tag{40}$$
$$k_1 = k_2 = 0, \; w_1 = 0.05; \; w_2 = 3.0.$$

Figure 11a,b show that in the case of a linear trajectory, the controller tries to follow it but the vehicle's movement is oscillatory. Such motion is also evidenced by the history of velocity, the force and torque, and the kinetic energy (Figure 11c–e). The controller is operating at maximum values (Figure 11d), which are changing rapidly, which would probably lead to its destruction. In summary, the controller fails even under simplifying assumptions. For a complex trajectory, this controller also fails under the assumed operating conditions.

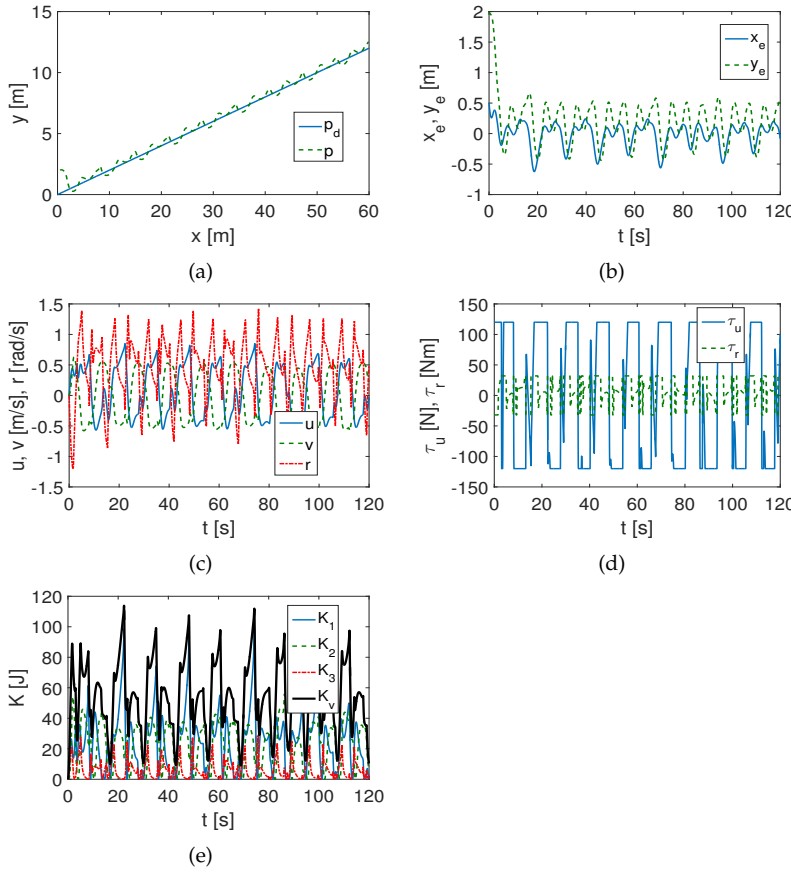

**Figure 11.** Results for EZY2016 controller and linear trajectory: (**a**) desired and realized trajectory; (**b**) position errors; (**c**) velocities; (**d**) applied force and torque; (**e**) kinetic energy time history.

### 4.3.4. Global Integral Sliding Mode Control

The trajectory tracking control scheme developed in [10] (denoted here as JGY2018) consists of a kinematic controller designed based on the backstepping method and a dynamic controller using the global integral sliding mode control. The equations of motion are of the form (10)–(12) but with the disturbance functions $f_u, f_v, f_r$ omitted. The modeling

errors are assumed to be differences between nominal and real values, for which limits are given. To test the C-Ranger vehicle, software was adapted from [57]. The following set of controller parameters was assumed:

$$\text{Linear trajectory } k_x = 20, \; k_\psi = 2, \; \lambda_1 = \lambda_2 = 0.1, \; \Gamma_1 = \Gamma_2 = 1.0, \tag{41}$$
$$\beta_1 = 0.5, \; \beta_2 = 10; \; W = 0.2,$$
$$\text{Complex trajectory } k_x = 20, \; k_\psi = 2, \; \lambda_1 = \lambda_2 = 0.1, \; \Gamma_1 = \Gamma_2 = 1.0, \tag{42}$$
$$\beta_1 = 0.5, \; \beta_2 = 10; \; W = 0.2.$$

Figure 12a,b show that linear trajectory tracking is possible with this controller. However, the position errors are large and the steady state is not obtained because the controller is constantly trying to overcome the presence of dynamic couplings. The time histories shown in Figure 12c–e, on the contrary, suggest that the controller is working properly, which is unfortunately misleading. Similar observations can be made from the results presented in Figure 13a–e for the complex trajectory.

In summary, the controller works in the case of dynamic coupling, but the tracking errors have large values that change and do not tend to a steady state under the assumed operating conditions.

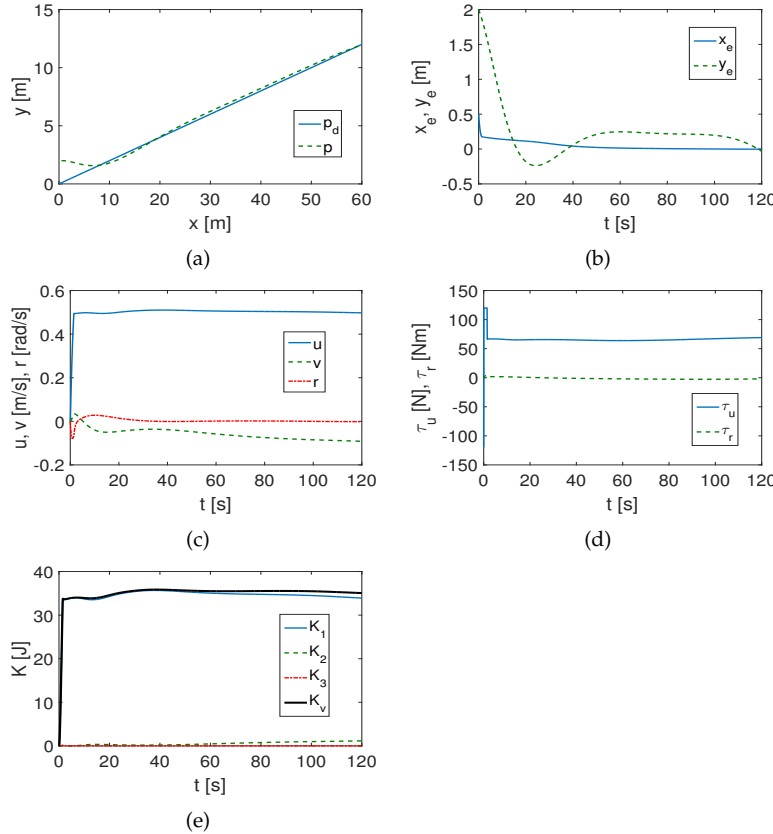

**Figure 12.** Results for JGY2018 controller and linear trajectory: (**a**) desired and realized trajectory; (**b**) position errors; (**c**) velocities; (**d**) applied force and torque; (**e**) kinetic energy time history.

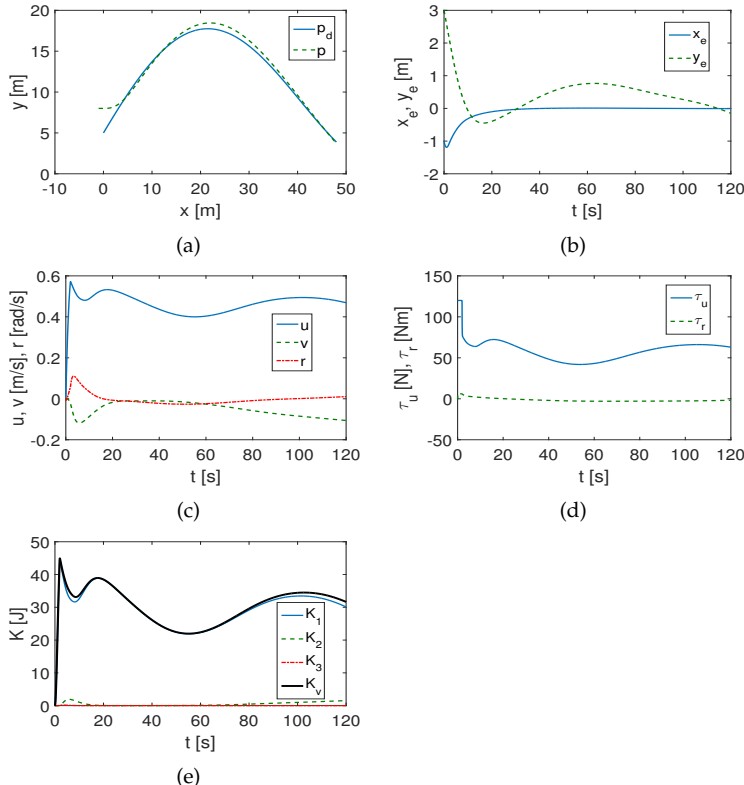

**Figure 13.** Results for JGY2018 controller and complex trajectory: (**a**) desired and realized trajectory; (**b**) position errors; (**c**) velocities; (**d**) applied force and torque; (**e**) kinetic energy time history.

### 4.3.5. Tracking Control by Modified Dynamic Inversion

The three methods developed in [24] (denoted here as YZ2018) were based on a modification of dynamic inversion methods. The equations of motion for which the control algorithms were designed were (10)–(12) but with the uncertainties model $f_u, f_v, f_r$ omitted and the quantities $d_{11}, d_{22}, d_{33}$ limited to constant values only. The reference shows that the dynamic inversion method based on output redefinition (ORDI) was the most robust to model uncertainties. Therefore, it was used here for comparative testing. The simulation tests were conducted on the software from [58] after adaptation to the C-Ranger vehicle. The following controller parameters were applied:

$$\text{Linear trajectory } k_{31} = 40, \ k_{32} = 3, \ k_{33} = 40, \ k_{34} = 3, \ l_1 = 0.01; \tag{43}$$

$$\text{Complex trajectory } k_{31} = 50, \ k_{32} = 0.7, \ k_{33} = 50, \ k_{34} = 0.7, \ l_1 = 0.01. \tag{44}$$

It can be seen from Figure 14a that for a linear trajectory and only for nominal parameters (exactly known and assumptions that simplify the model), the tracking task is possible. However, when couplings are included in the model, the algorithm completely fails, as shown in Figure 14b,c. The time histories of the quantities in Figure 14d–f are completely misleading as they do not indicate controller ineffectiveness. When the complex trajectory was used, the results in Figure 15a–f were the same as before; that is, the controller was inefficient and failed in the case of couplings.

The conclusion is that the controller is not useful for trajectory tracking when dynamic couplings are present.

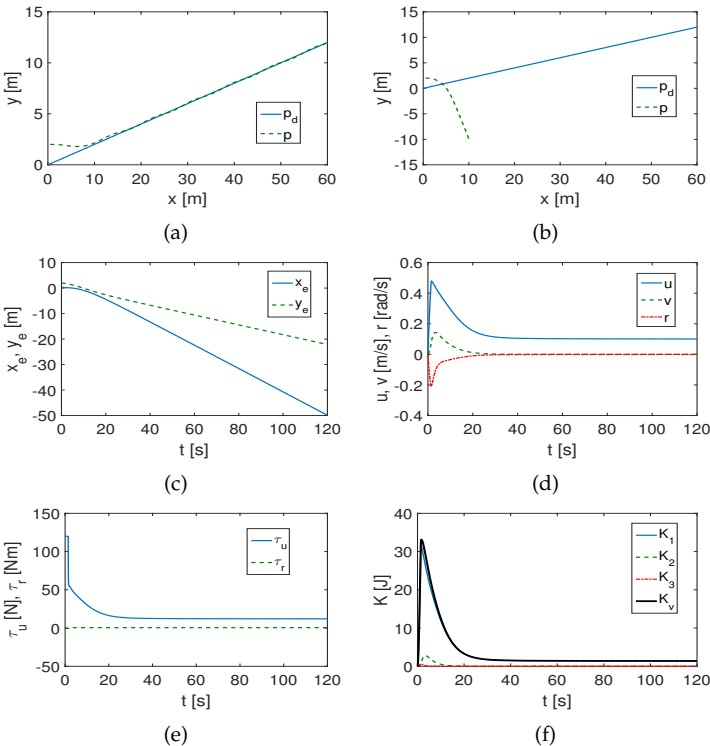

**Figure 14.** Results for YZ2018 controller and linear trajectory: (**a**) desired and realized trajectory for nominal case; (**b**) desired and realized trajectory for couplings; (**c**) position errors for couplings; (**d**) velocities for couplings; (**e**) applied force and torque for couplings; (**f**) kinetic energy time history for couplings.

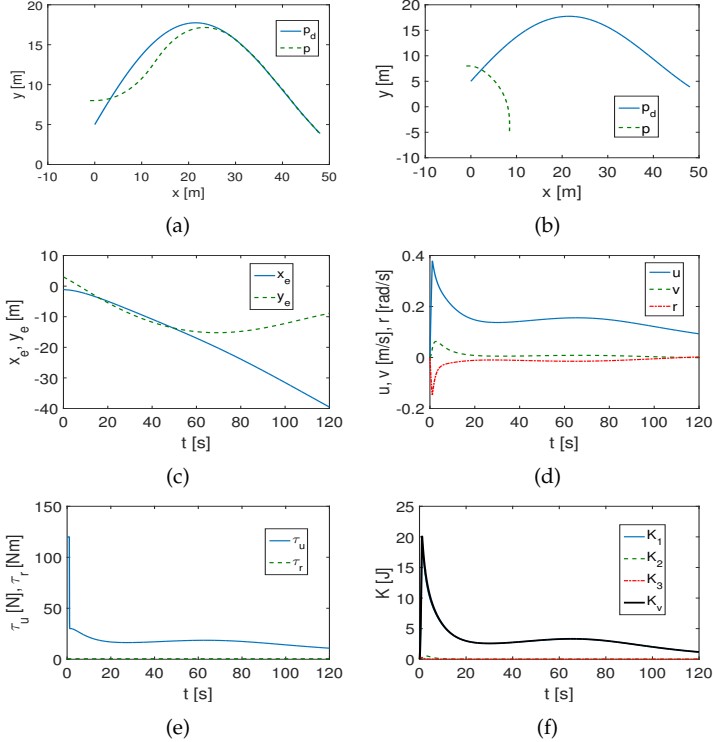

**Figure 15.** Results for YZ2018 controller complex trajectory: (**a**) desired and realized trajectory for nominal case; (**b**) desired and realized trajectory for couplings; (**c**) position errors for couplings; (**d**) velocities for couplings; (**e**) applied force and torque for couplings; (**f**) kinetic energy time history for couplings.

4.3.6. Analysis of Results Based on Indexes

Numerical results based on the assumed indexes are shown in Table 2. The values for the algorithm that failed completely (YZ) and the algorithm that gave unacceptable results for the complex trajectory (EZY) are omitted.

*Linear trajectory*. The results suggest some robustness to model asymmetry for XWQ, SZQZ, JGY while not necessarily for EZY (for example, indexes MIAC $\tau_u$ and MIAC $\tau_r$ indicate a problem with the algorithm's robustness). Consistency with the time history of the corresponding signals can only be observed for SZQZ and XWQ, but for JGY, this is not the case because no steady-state results were obtained.

*Complex trajectory*. According to the indexes, the XWQ and JGY algorithms may be robust but not SZQZ (indexes MIAC $\tau_u$ and MIAC $\tau_r$ suggest a problem with the algorithm's robustness). While the XWQ algorithm was able to cope with the situation of a shifting center of mass (although not without difficulties), for JGY, there is the same problem as before (no steady state at the assumed time).

**Table 2.** Performance for tested methods and for different trajectories.

| | | | Linear | | | | | Complex | |
|---|---|---|---|---|---|---|---|---|---|
| **Index** | **QV** | **XWQ** | **SZQZ** | **EZY** | **JGY** | **QV** | **XWQ** | **SZQZ** | **JGY** |
| MIA $x_e$ | 0.0125 | 0.2741 | 0.0146 | 0.1555 | 0.0444 | 0.0227 | 0.3323 | 0.6404 | 0.0851 |
| MIA $y_e$ | 0.1708 | 0.1828 | 0.1703 | 0.3334 | 0.2796 | 0.2028 | 0.2312 | 0.4626 | 0.4779 |
| mean $x_e$ | −0.0044 | −0.2701 | −0.0054 | −0.0136 | 0.0441 | −0.0154 | 0.1329 | −0.0078 | −0.0804 |
| std $x_e$ | 0.0333 | 0.1648 | 0.0449 | 0.2021 | 0.0587 | 0.1033 | 0.3826 | 0.7807 | 0.2185 |
| mean $y_e$ | 0.1711 | 0.0624 | 0.1707 | 0.2114 | 0.2260 | 0.2017 | 0.0597 | 0.1168 | 0.3725 |
| std $y_e$ | 0.3837 | 0.3701 | 0.3804 | 0.3828 | 0.3907 | 0.5078 | 0.4513 | 0.5920 | 0.5127 |
| RMS $\|\|e\|\|$ | 0.4204 | 0.4900 | 0.4184 | 0.4809 | 0.4564 | 0.5544 | 0.6076 | 0.9856 | 0.6736 |
| MIAC $\tau_u$ | 66.214 | 68.123 | 66.882 | 105.04 | 66.338 | 57.396 | 89.644 | 117.27 | 58.949 |
| MIAC $\tau_r$ | 1.9068 | 2.9175 | 2.2082 | 19.872 | 1.7906 | 2.2887 | 7.1284 | 31.573 | 2.2931 |
| mean $K$ | 34.963 | 36.603 | 34.988 | 53.865 | 34.987 | 29.461 | 32.024 | 48.826 | 30.425 |

*4.4. Discussion of Results*

The results obtained under the assumed testing conditions can be summarized in a series of points.

1.  The best results were generally obtained when using a control scheme designed for vehicles described by a fully asymmetric model in the horizontal plane, that is, taking into account dynamic couplings. This is not an unexpected result because these types of algorithms are designed precisely for the control of asymmetric vehicles.
2.  Using only indexes, but no time history, can be misleading and is certainly not sufficient to test the suitability of the algorithm. Both tests are essential.
3.  Analysis of time histories and indexes show that the XWQ2015 controller is the most robust to vehicle asymmetry. The SZQZ2018 algorithm worked satisfactorily only for the linear trajectory. The JGY2018 control algorithm managed in the given situation, but the results were not satisfactory. The EZY2016 and YZ2018 algorithms proved to be completely unable to cope with such a situation.

Based on the tests, some conclusions can also be drawn about the design of control algorithms.

1.  Designing control strategies for vehicle models that take into account the possibility that the geometric center is not identically located to the center of mass seems as expedient as possible. First of all, such a situation could potentially occur for a number of reasons (e.g., lack of precisely known parameters, displacement of cargo or mounted equipment). Second, such a control scheme can be simplified and applied to a fully symmetrical vehicle.
2.  When designing a controller for a fully symmetric vehicle (a model with a diagonal inertia matrix), it is important to verify that it will also work if the center of mass is

shifted slightly. Often a complex mathematical theory is used in control algorithms and the results are satisfactory only under strictly assumed conditions. On the other hand, if the situation in question occurs, an algorithm that is not robust to the asymmetry of the vehicle model will fail and will have only theoretical significance but be without much value from a technical point of view. In such a situation, experimental studies should be carried out to answer the question to what extent the algorithm is useful.

3. Test studies using simulation appear to be necessary to evaluate the suitability of the controller before implementing it in a real object. They can also be used when an experiment is not planned but one wants to explore some control idea. In this way, one can avoid the costly expense of an experiment that may not yield the expected results or prove difficult to implement.

*4.5. Advantages and Disadvantages of Control Schemes and Their Possible Applications*

The main advantage of the QV scheme is that it takes into account shifting of the center of mass, which makes it possible to effectively control the vehicle when this situation occurs. The controller's performance is acceptable for both linear and more complex trajectory tracking. The disadvantage, however, is the greater complexity of the controller equations due to the more complicated dynamic model.

The benefit of control scheme XWQ2015 is that it is guaranteed to be effective in tracking different trajectories, as shown in [8], but provided the vehicle dynamics model is diagonal. Unfortunately, when the center of mass is displaced, tracking errors are large, which is a disadvantage of this controller.

The SZQZ2018 algorithm has the advantage that it is effective for a model with a diagonal inertia matrix and in the case of shifted center of mass it can be used to track a linear trajectory. When the trajectory has a complex form, the controller fails which is its disadvantage.

The benefit of the EZY2016 algorithm, as shown in [3], is that it can track different trajectories, but in the presence of little disturbance. If the center of mass is displaced, the control scheme fails, which is a disadvantage.

The advantage of the JGY2018 controller, as illustrated in [10], is its ability to track different trajectories when model parameters are disturbed. However, if there are external disturbances and, in addition, the center of mass is displaced, the algorithm will not track the desired trajectory, which is its disadvantage.

In [24], it was demonstrated that the benefit of the YZ2018 controller is effective trajectory tracking with uncertain model parameters. But the assumed model does not include components to account for other disturbances, which is a disadvantage. In addition, when the center of mass is shifted, satisfactory results, as were presented in this paper, cannot be guaranteed.

As a result of the tests, it was found that the control scheme designed for the model with a symmetric inertia matrix leads to satisfactory results. On the other hand, when the algorithm is designed for a model with a diagonal inertia matrix, and there will be shifts in the center of mass, acceptable performance can be obtained only under special conditions. It also depends on the method or combination of methods used in the controller.

If a simplified model (with a diagonal inertia matrix) is used, the control scheme should be based on methods that do not have a complex mathematical form. Otherwise, it is difficult to expect effective results when applied to an asymmetric vehicle. When the control method for a simpler model is not mathematically complex, it can be extended to a more complicated model. Examples of such methods here are SZQZ2018 and JGY2018, which can be modified to be suitable for a model containing a symmetric inertia matrix. Although in its original form the latter algorithm did not work successfully, the control concept used can be extended. Modifying these two algorithms, however, will increase the complexity of the controller but expand its usefulness.

A more general statement is that if a control algorithm does not work well in simulation tests and cannot be easily modified to fit a more complicated model (thus increasing its numerical complexity), it is unlikely that it can be expected to give acceptable results in experimental investigations. Confirmation of this assumption can be found in the fact that, for example, in [30,31,38,39], satisfactory simulation results were first obtained for the proposed control strategies, and then their effectiveness was confirmed in a real experiment.

In conclusion, it can be stated that unless the diagonal model of the vehicle is replaced by a model with an asymmetric inertia matrix, only some of the proposed algorithms will guarantee acceptable tracking results, and this is provided that the trajectory is linear or close to it. Obviously, a more complex model will also lead to more complex controller equations.

## 5. Conclusions

This paper addresses the problem of testing the robustness of control algorithms when a controller designed for a fully symmetric vehicle is used in a situation of small asymmetry (a few percent). The method involves establishing a basic algorithm (designed for asymmetric vehicles) and then verifying, based on the assumed procedure and indexes of the results obtained for other algorithms (suitable for symmetric vehicles) and determining whether they are robust in such a situation. Tests using the proposed method show that the problem exists, although it is usually ignored. This leads to control schemes that have value from a mathematical point of view while their practical value can be easily questioned. Of the five selected algorithms, only one showed robustness to the vehicle asymmetry considered in the mathematical model, two only managed the tracking task to a very limited extent, and the remaining two failed altogether. Therefore, it can be concluded that the problem exists and is solved only in a small number of cases. For this reason, it seems necessary to conduct research in the area under discussion, which concerns the design of controllers for predetermined models of marine vehicles.

**Funding:** The work was supported by Poznan University of Technology Grant No. 0211/SBAD/0123.

**Institutional Review Board Statement:** Not applicable.

**Informed Consent Statement:** Not applicable.

**Data Availability Statement:** The original contributions presented in the study are included in the article.

**Conflicts of Interest:** The author declares no conflicts of interest.

## Appendix A

The matrices and vectors in Equations (1) and (2) are of the form:

$$
R(\psi) = \begin{bmatrix} \cos\psi & -\sin\psi & 0 \\ \sin\psi & \cos\psi & 0 \\ 0 & 0 & 1 \end{bmatrix}, \quad M = \begin{bmatrix} m_{11} & 0 & m_{13} \\ 0 & m_{22} & m_{23} \\ m_{13} & m_{23} & m_{33} \end{bmatrix},
$$

$$
C(v) = \begin{bmatrix} 0 & 0 & c_{13} \\ 0 & 0 & c_{23} \\ -c_{13} & -c_{23} & 0 \end{bmatrix}, \quad D(v) = \begin{bmatrix} d_{11} & 0 & d_{13} \\ 0 & d_{22} & d_{23} \\ d_{31} & d_{32} & d_{33} \end{bmatrix}, \quad f_{ed} = \begin{bmatrix} f_{u\,exd} \\ f_{v\,exd} \\ f_{r\,exd} \end{bmatrix}, \tag{A1}
$$

where:

$$
m_{11} = m - X_{\dot{u}}, \quad m_{13} = m_{31} = -m y_g, \quad m_{22} = m - Y_{\dot{v}},
$$
$$
m_{23} = m_{32} = m x_g, \quad m_{33} = J_z - N_{\dot{r}},
$$
$$
c_{13} = -m_{22}v - m_{23}r, \quad c_{23} = m_{11}u - m_{13}r,
$$
$$
d_{11} = X_u + X_{|u|u}|u|, \quad d_{13} = X_r + X_{|u|r}|u|, \quad d_{22} = Y_v + Y_{|v|v}|v|, \quad d_{23} = Y_r + Y_{|v|r}|v|,
$$
$$
d_{31} = N_u + N_{|r|u}|r|, \quad d_{32} = N_v + N_{|r|v}|r|, \quad d_{33} = N_r + N_{|r|r}|r|, \tag{A2}
$$

in which $m$ stands for mass, $J_z$ for inertia, and other symbols stand for inertia components, including added masses, Coriolis components and hydrodynamic damping terms (linear and quadratic drag), respectively.

The equations of motion replacing (3) and (4) are as follows:

$$\zeta_1 = u - \hat{\mathbb{P}}_{13}r, \tag{A3}$$

$$\zeta_2 = v - \hat{\mathbb{P}}_{23}r, \tag{A4}$$

$$\zeta_3 = r, \tag{A5}$$

$$\mathbb{A}_1\dot{\zeta}_1 = H_1(\nu) + \tau_u + f_u, \tag{A6}$$

$$\mathbb{A}_2\dot{\zeta}_2 = H_2(\nu) + f_v, \tag{A7}$$

$$\mathbb{A}_3\dot{\zeta}_3 = H_3(\nu) + \tau_{\zeta_3} + f_{\zeta_3}, \tag{A8}$$

with $\tau_{\zeta_3} = \hat{\mathbb{P}}_{13}\tau_u + \tau_r$, $\;f_{\zeta_3} = \hat{\mathbb{P}}_{13}f_u + \hat{\mathbb{P}}_{23}f_v + f_r$, and:

$$H_1(\nu) = (m_{22}v + m_{23}r)r - d_{11}u - d_{13}r, \tag{A9}$$

$$H_2(\nu) = (-m_{11}u + m_{13}r)r - d_{22}v - d_{23}r, \tag{A10}$$

$$H_3(\nu) = -(m_{22}v + m_{23}r)u + (m_{11}u - m_{13}r)v + \hat{\mathbb{P}}_{13}(m_{22}v + m_{23}r)r - \hat{\mathbb{P}}_{23}(m_{11}u - m_{13}r)r$$
$$- (\hat{\mathbb{P}}_{13}d_{11} + d_{31})u - (\hat{\mathbb{P}}_{23}d_{22} + d_{32})v - (\hat{\mathbb{P}}_{13}d_{13} + \hat{\mathbb{P}}_{23}d_{23} + d_{33})r. \tag{A11}$$

For simplicity, the symbols are introduced $H_1 = H_1(\nu)$, $H_2 = H_2(\nu)$, and $H_3 = H_3(\nu)$.

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
