# Peer review of "Model Simplification for Asymmetric Marine Vehicles in Horizontal Motion—Verification of Selected Tracking Control Algorithms"

_electronics, doi:10.3390/electronics13101820_

Round 1

Reviewer 1 Report

Comments and Suggestions for Authors

This paper primarily investigates the feasibility of simplifying models for asymmetric marine vehicles in horizontal motion, and the performance of various tracking control algorithms implemented on such simplified models. The topic of the paper has high practical application value; research on control systems for asymmetric marine vehicles can advance technology in related fields. The author proposes a new control algorithm theoretically and validates its effectiveness on asymmetric models through simulation tests, demonstrating a certain level of innovation. The paper is well-structured with rigorous logic in each section, and the experimental design and results analysis are thorough, effectively supporting the author's conclusions.

Here are some review comments for the paper:

  1. Although the article compares several control algorithms, the explanation of specific scenarios and limitations applicable to each algorithm is not detailed enough. It is recommended that the author adds a comparison of the advantages and disadvantages of each algorithm in the discussion section, as well as the specific scenarios where they are applicable.
  2. The paper frequently discusses the impacts of model simplification. It is suggested that the author further discuss how to balance the need for model simplification with the complexity of control algorithms in practical applications, and how this balance specifically affects system performance.
  3. The clarity of some figures and charts in the article is low, especially the key flowcharts of control algorithms and performance comparison charts. It is recommended to redraw these to ensure the clarity and readability of the information in the diagrams.
  4. Some theoretical derivations in the article are complex and may not be easily understood by non-expert readers. It is suggested that the author provides more detailed derivations or background knowledge in an appendix or supplementary materials.

Overall, this paper presents certain innovative and practical values and can provide useful references for research on control algorithms for asymmetric marine vehicles. However, there is room for improvement in detail and clarity in some parts of the paper. It is recommended for acceptance after major revisions.

Comments on the Quality of English Language

No

Author Response

Thank you for your review.  In a separate PDF file I am sending answers to the problems indicated.

Reviewer 2 Report

Comments and Suggestions for Authors

The paper subject is a very interesting one and a very actual due the fast development of the marine drones. The list of references is a relevant one, with some classical titles in the control area and in the proposed subject, but about three quarters of the titles are from the last 5 years, reflecting the actuality of the subject.

The introductory part is good considering the cited works, but it is a very short and concentrated one. It is important to have a more detailed introduction.

The theoretical part is very good emphasized for scientific point of view. Unfortunately, the goal related to the asymmetry effects is too much dissipated in the scientific consistency of the chapters 2 and 3 and this is not helping the average readers.

Chapter 4 is trying to cover all the missing elements. It presents the considered algorithms, the simulation results (some preliminary data on the simulation setup may help) and also some comparisons and specific conclusions between these results. The general conclusions are presented in chapter 5.

I suggest to emphasize the chapters 1-3 in the direction to prepare better (and in a more accessible language for the average readers) the simulations and the rsults of chapter 4.

Author Response

(The authors gave the same response as above.)

Round 2

Reviewer 1 Report

Comments and Suggestions for Authors

The article has been appropriately modified, and the author has also addressed my concerns. I believe this article can be accepted.

Comments on the Quality of English Language

No

Reviewer 2 Report

Comments and Suggestions for Authors

I appreciate the author's effort to reorganize the paper to increase its readability. 

I consider the paper as a necessary one in the present context of knowledge evolution in the topic.